# Prognostic Value of Mid-Regional Proadrenomedullin Sampled at Presentation and after 72 Hours in Septic Patients Presenting to the Emergency Department: An Observational Two-Center Study

**DOI:** 10.3390/biomedicines10030719

**Published:** 2022-03-19

**Authors:** Paolo Bima, Giorgia Montrucchio, Valeria Caramello, Francesca Rumbolo, Stefania Dutto, Sarah Boasso, Anita Ferraro, Luca Brazzi, Enrico Lupia, Adriana Boccuzzi, Giulio Mengozzi, Fulvio Morello, Stefania Battista

**Affiliations:** 1S.C. Medicina d’Urgenza U (MECAU), Ospedale Molinette, A.O.U Città della Salute e della Scienza di Torino, 10126 Torino, Italy; paolo.bima@edu.unito.it (P.B.); stefania.dutto@unito.it (S.D.); sarah.boasso@unito.it (S.B.); enrico.lupia@unito.it (E.L.); sbattista@cittadellasalute.to.it (S.B.); 2Scuola di Specializzazione in Medicina di Emergenza e Urgenza, 10126 Torino, Italy; 3S.C. Anestesia e Rianimazione 1U, Dipartimento di Anestesia, Terapia Intensiva ed Emergenza, Ospedale Molinette, A.O.U Città della Salute e della Scienza di Torino, 10126 Torino, Italy; giorgiagiuseppina.montrucchio@unito.it (G.M.); luca.brazzi@unito.it (L.B.); 4Dipartimento di Scienze Chirurgiche, Università degli Studi di Torino, 10126 Torino, Italy; anita.ferraro@unito.it (A.F.); adriana.boccuzzi@unito.it (A.B.); 5S.C. Medicina d’Urgenza, A.O.U. San Luigi Gonzaga, 10043 Orbassano, Italy; v.caramello@sanluigi.piemonte.it; 6S.C. Biochimica Clinica, A.O.U Città Della Salute e Della Scienza, 10126 Torino, Italy; francesca.rumbolo@unito.it (F.R.); giulio.mengozzi@unito.it (G.M.); 7Dipartimento di Scienze Mediche, Università degli Studi di Torino, C.so Bramante 88, 10126 Torino, Italy

**Keywords:** sepsis, prognosis, biomarker, mid-regional proadrenomedullin, emergency department

## Abstract

The prognostic value of mid-regional proADM (MR-proADM) in septic patients presenting to the emergency department (ED) is not well established. In this prospective observational study enrolling septic patients evaluated in two EDs, MR-proADM was measured at arrival (t0) and after 72 h (t72). MR-proADM_%change_ was calculated as follows: (MR-proADM_t72h_ − MR-proADM_t0_)/MR-proADM_t0_. In total, 147 patients were included in the study, including 109 with a final diagnosis of sepsis and 38 with septic shock, according to the Sepsis-3 criteria. The overall 28-day mortality (outcome) rate was 12.9%. The AUC for outcome prognostication was 0.66 (95% CI 0.51–0.80) for MR-proADM_t0_, 0.77 (95% CI 0.63–0.92) for MR-proADM_t72_ and 0.74 (95% CI 0.64–0.84) for MR-proADM_%change_. MR-proADM_t0_ ≥ 2.78 nmol/L, MR-proADM_t72_ ≥ 2.7 nmol/L and MR-proADM_%change_ ≥ −15.2% showed statistically significant log-rank test results and sensitivity/specificity of 81/65%, 69/80% and 75/70% respectively. In regression analysis, MR-proADM_%change_ was a significant outcome predictor both in univariate and multivariate analysis, after adjustment for age, SOFA and APACHEII scores, providing up to 80% of added prognostic value. In conclusion, time trends of MR-proADM may provide additional insights for patient risk stratification over single sampling. MR-proADM levels sampled both at presentation and after 72 h predicted 28-day survival in septic patients presenting to the ED.

## 1. Background

Sepsis and septic shock are life-threatening conditions arising from a dysregulated immune response to an infection, [1] with high incidence and high fatality rate, despite progress in its recognition and its management [2]. A key factor to improve mortality, beyond the early identification and its proper management in the first hours, is a better prognostic stratification in order to assign the most correct allocation of each patient and monitor treatment response to adjust or upgrade treatments as soon as failure is suspected.

Adrenomedullin (ADM) is a multipotent regulatory peptide with several biological activities—vasodilator, positive inotropic, diuretic, natriuretic and bronchodilator—that are widely expressed throughout the body. It is upregulated by hypoxia, inflammatory cytokines, bacterial products and shear stress [3]. Adrenomedullin (ADM) is a peptide produced by multiple tissues in response to an infective stimulus, leading to a rapid increase in its plasma concentration, with apparently beneficial effects on the pathophysiology of sepsis. Due to the increased stability, the precursor of ADM, mid-regional proADM (MR-proADM), is clinically measured instead [4].

Previous studies have identified MR-proADM as a potential candidate biomarker to predict mortality and treatment response in septic patients [5,6,7,8,9]. However, appropriate cut-off values and sampling time are unknown [10,11]. As for other biomarkers [12,13,14], its predictive value might improve using serial measurements, owing to biological reasons and dependence on treatment response. The aim of the present study was the assessment of the ability of MR-proADM, measured upon ED arrival and after 72 h, to predict 28-day mortality and its additional prognostic value on top of commonly used clinical biomarkers and scores in patients with sepsis or septic shock presenting to the ED.

## 2. Methods

### 2.1. Study Design

This prospective observational study was conducted from July 2019 to January 2020 in two EDs of university hospitals in Piedmont, Italy (A.O.U. Città della Salute e della Scienza—Molinette Hospital, Torino and A.O.U. San Luigi Gonzaga, Orbassano). The study was approved by the local ethics committee (protocol number 0061165, approved on the 17 June 2019) and was conducted according to the Declaration of Helsinki. Informed consent was obtained by all the participants. The management of the included patients was done in accordance with local and international guidelines, irrespective of their participation in the study. The attending physician was blinded to the MR-proADM results, but not to the results of the other requested laboratory tests, including other common biomarkers used in clinical practice such as C-reactive protein (CRP), procalcitonin (PCT) and lactate.

### 2.2. Inclusion and Exclusion Criteria

Patients ≥ 18 years old were included if sepsis or septic shock was diagnosed by the attending physician on the bases of history, clinical evaluation and routine laboratory tests obtained during the index ED visit. Patients were excluded if <18 years old or if they refused to sign the informed consent. To allow for immediate centrifugation and freezing of plasma samples in the laboratory, enrolment was limited to daytime, from 8 a.m. to 6 p.m., Monday to Friday.

### 2.3. Outcome and Study Plan

The outcome of the study was 28-day mortality. We planned to evaluate the ability of MR-proADM, measured at arrival and after 72 h, to predict the outcome either as a single measure or as a trend and the additive prognostic ability on top of the SOFA [15] and APACHEII scores [16].

### 2.4. Clinical Management

Upon admission to the ED, the patient’s demographic data and clinical history were collected. Patients underwent assessments of clinical (blood pressure, heart rate, respiratory rate, oxygen saturation and body temperature) and laboratory parameters (full blood cell count, C-reactive protein, procalcitonin, renal function, electrolytes and arterial blood gas analysis), which were ordered by the attending physician on a case-by-case ratio. When necessary, analyses of blood culture, sputum, urine, bronchial aspirate or bronchoalveolar samples were also performed. The infection source, as well as the requirement of vasoactive drugs with the development of septic shock, were registered according to the European Centre for Disease Prevention and Control (ECDC) current definitions [17]. Clinical severity was determined by calculating SOFA and APACHE II scores.

### 2.5. Biomarker Measurement

MR-proADM was sampled at arrival, and after 72 h, the B.R.A.H.M.S. KRYPTOR compact PLUS (Thermo Fisher Scientific, Hennigsdorf, Germany) automated method using the Time-Resolved Amplified Cryptate Emission (TRACE) technique was conducted, using plasma aliquots from EDTA-containing tubes, which were centrifugated at 4000× *g* for five minutes and then immediately frozen and stored at −80 °C. The detection limit of the assay was 0.05 nmol/L, while intra- and inter-assay coefficients of variation were under 4% and 11%, respectively.

The time trend of MR-proADM levels, defined as the percent change between 72 h and arrival, was calculated as follows: MR-proADM_%change_ = (MR-proADM_t72h_ − MR-proADM_t0_)/MR-proADM_t0_; positive values of MR-proADM_%change_ indicate an increasing trend, and negative values indicate a decreasing trend.

### 2.6. Case Adjudication

Two expert study investigators independently reviewed the hospital charts, including all lab and cultural results (excluding MR-proADM levels), imaging and surgical and pathology data in order to adjudicate all cases according to the Sepsis-3 criteria. In case of discordance, discussion between the two investigators and other authors was planned. Patients who did not meet the criteria for sepsis or septic shock definition were not further analyzed.

### 2.7. Sample Size Calculation

Using a power of 80%, a one-tailed alpha error of 0.05 and a sampling ratio of 1:1, we estimated that 105 patients were needed to detect an absolute difference of 20% [6] in 28-day mortality using proADM (mortality 1: 10%, mortality 2: 30%).

### 2.8. Statistical Analysis

Continuous variables were expressed as mean ± standard deviation (SD) if normally distributed, otherwise as median and interquartile range (IQR) and were compared using the Mann–Whitney U test. Categorical variables were expressed as absolute number (percentage) and compared with the χ^2^ or the Fisher’s exact test as appropriate.

The prognostic ability of MR-proADM, SOFA and APACHE II scores was assessed with receiver operating characteristic (ROC) curves and their area under the curve (AUC). The ROC curves were compared using the DeLong’s test. The Youden index was used to find the optimal cut-off values, defined as the maximum value of sensitivity + specificity − 1.

Logistic regression analysis was performed to assess if MR-proADM was an independent predictor of mortality, either alone (univariate) or adjusted for age, SOFA and APACHEII scores (multivariate). Multiple models were planned, including the use of MR-proADM as a continuous variable and a binary variable after dichotomization with previously or newly found cut-offs. The odds ratio (OR) and its 95% confidence interval (95% CI) were computed. In case of non-linearity of the relationship between the predictor and outcome, cubic spline modeling of the continuous variable was pursued [18]. To assess the simultaneous effects of the two continuous predictors, MR-proADM_t0_ and MR-proADM_t72h_, on the probability of 28-day mortality, a contour plot was drawn. To further verify that MR-proADM had additional prognostic value over the confounding variables, the obtained logistic regression model for each confounding variable alone was compared using a likelihood ratio (LR) test. If the LR test was statistically significant, the fraction of added prognostic value of MR-proADM to a model only with the confounding variable was calculated as 1 − LR_without-MR-proADM_/LR_with-MR-proADM_ [19,20]. Survival analysis was performed using the Kaplan–Meier estimator for the cut-off values found with ROC analysis or previously published cut-offs and the curves were compared using the log-rank test. Sensitivity and specificity were calculated for the best cut-offs and compared using the exact binomial method [21]. *p*-values were considered significant if <0.05. The statistical analysis was performed using R (version 3.6.4).

## 3. Results

### 3.1. Patient Characteristics

In total, 147 patients meeting the Sepsis-3 criteria were included in the study. Their clinical characteristics are detailed in Table 1. The final adjudicated diagnosis was sepsis in N = 109 (74.1%) and septic shock in N = 38 (25.9%, Appendix A). Respiratory, urinary and gastrointestinal infections were the most represented. Overall, 19 (12.9%) patients died within 28 days from admission. Culture positivity and type of microbiological isolates were not statistically different between survivors and non-survivors.

The median SOFA and APACHEII scores were 6 (4–7) and 29 (23–37) in non-survivors vs. 4 (2–6) and 20 (15–24) in survivors (*p* = 0.03 and *p* < 0.001, respectively). With regards to treatment, 38 patients (25.9%) were treated with vasopressors, without differences between the groups. Oxygen therapy, non-invasive ventilation, mechanical ventilation and renal replacement therapy were more frequent among non-survivors.

At presentation, the median MR-proADM_t0_ was 3.19 (1.36–9.37) nmol/L in non-survivors and 1.79 (1.07–3.78) nmol/L in survivors (*p* = 0.03). After 72 h, the median MR-proADM_t72h_ was 4.31 (2.00–8.86) nmol/L in non-survivors and 1.29 (0.82–2.30) nmol/L in survivors (*p* < 0.001, Figure 1A). The median MR-proADM_%change_ was 2% (−14–7) in non-survivors and −28% (−55–[−8]) in survivors (*p* = 0.002, Figure 1B).

### 3.2. Survival Analysis

The contour plot showing the simultaneous effects of MR-proADM_t0_ and MR-proADM_t72h_ on 28-day mortality is shown in Figure 2A. Figure 2B shows the ROC curves for outcome prognostication of study biomarkers and other clinically relevant variables. The AUC was 0.66 (95% CI 0.51–0.80) for MR-proADM_t0_, 0.77 (95% CI 0.63–0.92) for MR-proADM_t72_, 0.74 (95% CI 0.64–0.84) for MR-proADM_%change_, 0.57 (95% CI 0.43–0.72) for PCT_t0_, 0.57 (95% CI 0.43–0.72) for CRP_t0_ and 0.65 (95% CI 0.49–0.82) for lactate_t0_. No difference was found at the DeLong’s test in comparison to SOFA score or MR-proADM_t0_. The AUC for MR-proADM_t72_ was significantly higher than the AUC for PCT_t0_ (*p* = 0.03). The Youden’s index was 2.78 nmol/L for MR-proADM_t0_, 2.7 nmol/L for MR-proADM_t72_ and −15.2%. for MR-proADM_%change_.

Kaplan–Meier survival curves were computed for MR-proADM_t0_ (cut-off 2.78 nmol/L), MR-proADM_t72_ (cut-off 2.7 nmol/L) and MR-proADM_%change_ (cut-off −15.2%). All were able to differentiate prognosis with a statistically significant log-rank test (*p* = 0.003, *p* < 0.001 and *p* < 0.001, respectively, Figure 3A,B).

### 3.3. Subgroup Analysis—Septic Shock

Septic shock was diagnosed in 38 patients (Appendix A), of whom 8 (21.1%) died. Median MR-proADM_t0_ was 7.83 (2.99–12.2) nmol/L in non-survivors and 4.35 (1.86–7.51) nmol/L in survivors (*p* = 0.25); the median MR-proADM_t72_ and MR-proADM_%change_ were significantly higher in non-survivors (7.24 (2.93–11.65) nmol/L and 4% (−14–21)) than in survivors (2.31 (1.35–4.50) nmol/L and −21% (−55–4), *p* = 0.03 and *p* = 0.04, respectively). In ROC curve analysis (Appendix A), the AUCs of MR-proADM_t0_, MR-proADM_t72_ and MR-proADM_%change_ were 0.64 (95% CI 0.39–0.89), 0.76 (95% CI 0.50–1.0) and 0.74 (95% CI 0.58–0.91), respectively.

### 3.4. Subgroup Analysis—Pulmonary Infective Focus

The most common infective focus in our cohort was the lung. Among 57 patients, 10 (17.5%) had pulmonary infective focus. Median MR-proADM_t0_ was 2.07 (0.96–3.96) nmol/L in non-survivors and 1.56 (1.07–2.53) nmol/L in survivors (*p* = 0.6); median MR-proADM_t72_ was 2.89 (1.10–4.40) nmol/L in non-survivors and 1.06 (0.69–1.57) nmol/L in survivors (*p* = 0.07); median MR-proADM_%change_ was 3% (−13–12) in non-survivors and −28% (−52–(−17)) in survivors (*p* = 0.01). In ROC curve analysis (Appendix A), the AUCs of MR-proADM_t0_, MR-proADM_t72_ and MR-proADM_%change_ were 0.56 (95% CI 0.32–0.79), 0.70 (95% CI 0.44–0.96) and 0.78 (95% CI 0.62–0.93), respectively.

Prognostication accuracy data of MR-proADM are summarized in Table 2. Logistic regression analysis (Appendix A) was used to uncover any additional prognostic value of MR-proADM over clinical data, summarized by age, SOFA and APACHEII scores. Only MR-proADM_%change_ was found as an independent mortality predictor both in the univariate model and after adjustment for all clinical variables, both as a continuous variable and as a dichotomic variable (cut-off −15.2%). MR-proADM_%change_ provided added prognostic value to all clinical variables, both as a continuous or dichotomic variable (Figure 4 and Appendix A).

A bivariate logistic regression model was then obtained with MR-proADM_t72_ and MR-proADM_%change_, representing variables with the best prognostic ability. When modeled as continuous variables, MR-proADM_t72_ had an OR of 2.2 (95% CI 1.4–3.6) and MR-proADM_%change_ had an OR of 20.1 (95% CI 1.2–330), the latter providing 42% of added prognostic value. When dichotomized at the respective cut-offs, MR-proADM_t72_ had an OR of 4.6 (95% CI 1.4–14.7) and MR-proADM_%change_ had an OR of 4.9 (95% CI 1.4–17.3), the latter providing 37% of added prognostic value.

Within patients with septic shock, at logistic regression analysis, only MR-proADM_t72_ (continuous variable) and MR-proADM_%change_ ≥ −15.2% were statistically significant in the univariate and multivariate models, yielding an additional prognostic value of 19.9% and 37.4%–42.3%, respectively (Appendix A).

Within patients with a pulmonary infective focus, at logistic regression analysis, only MR-proADM_%change_ ≥ −15.2% was statistically significant in the univariate and multivariate models, yielding an additional prognostic value of 37.4%–87.0% (Appendix A).

## 4. Discussion

This study highlights the prognostic potential of MR-proADM, measured both at arrival and after 72 h, in predicting 28-day mortality in septic patients presenting to the ED. MR-proADM at presentation showed reasonably high sensitivity and specificity in predicting mortality but did not provide additional prognostic discrimination over SOFA and APACHEII scores, in contrast to previous studies [5,6,7,8]. On the other hand, MR-proADM measured after 72 h, allowing for the calculation of MR-proADM relative change compared to the presentation level, provided additional prognostication ability over the clinical variables alone.

The best cut-off for MR-proADM_t0_ found in this cohort was consistent with a previous larger randomized trial [6] conducted in ICU patients, possibly indicating that values of MR-proADM_t0_ ≥ 2.75 nmol/L also have a good prognostic performance in patients presenting to the ED. In a previous study conducted in the ED, the optimal cut-off was ≥1.54 nmol/L [7], likely due to differences in case mix and mortality rates. Indeed, evidence has been provided that infective focus and disease severity may play a role in MR-proADM levels [5,6,9]. Taken together, available data infer that the MR-proADM level at ED presentation might be evaluated by physicians in conjunction with clinical scores for risk-stratification of septic patients. Nonetheless, proof of additional value of MR-proADM over clinical evaluation alone remains uncertain.

To the best of our knowledge, this is the first study investigating the prognostic ability of the MR-proADM 3-day trend in sepsis. When measured 72 h after ED evaluation, MR-proADM_t72_ provided similar sensitivity but increased specificity over MR-proADM_t0_, indicating that persistently high MR-proADM levels identify patients at higher mortality risk, requiring the utmost clinical efforts. Furthermore, the relative change in MR-proADM levels emerged as the key additional prognostic element over clinical scores. In a previous study [6], MR-proADM concentrations >2.25 nmol/L at day 1, 4, 7 and 10 predicted a higher 28-day mortality, although delta changes in MR-proADM were not found as mortality predictors. The importance of serial measurements over time of prognostic biomarkers has been already highlighted for procalcitonin (PCT) in septic patients [12] and for MR-pro-ADM in lower respiratory tract infections [11] and in COVID-19 [10]. Potential advantages of serial measurements may derive from underlying biological mechanisms and from capacity to discriminate treatment responses. In the present study, we used a 72 h time interval for re-sampling in accordance with previous literature as well as with technical and logistic issues, but we cannot exclude that different time points may provide additional or more relevant information [10,11]. Future studies are needed to establish the most appropriate timing for serial measurements and their costs and cost effectiveness. Currently, in our institution a MR-proADM test costs around twice the cost of a PCT test.

Our study has relevant strengths: blinding to MR-proADM results, strict adherence to the Sepsis-3 criteria, independent central case adjudication and adjustment for multiple confounders in statistical analysis. However, it also has limitations, as follows: (1) the sample size was smaller in comparison to previous studies; (2) 14.3% of MR-proADM_t72_ values were missing; (3) SOFA and APACHE II scores were calculated only at presentation, and not repeated at 72 h; (4) finally, the role of possible impact of confounding factors on endothelial damage, such as cardiovascular dysfunction and renal failure, cannot be clearly defined in this cohort of patients.

## 5. Conclusions

The present study shows that MR-proADM levels have prognostic value for septic patients evaluated in the ED. Measurement of MR-proADM after 72 h, allowing for the calculation of a percent change, provided additional prognostic value over clinical data. Further studies are needed to confirm these results, defining the best timing for MR-proADM second draw and to explore the potential of MR-proADM-based diagnostic or treatment algorithms.

## Figures and Tables

**Figure 1 biomedicines-10-00719-f001:**
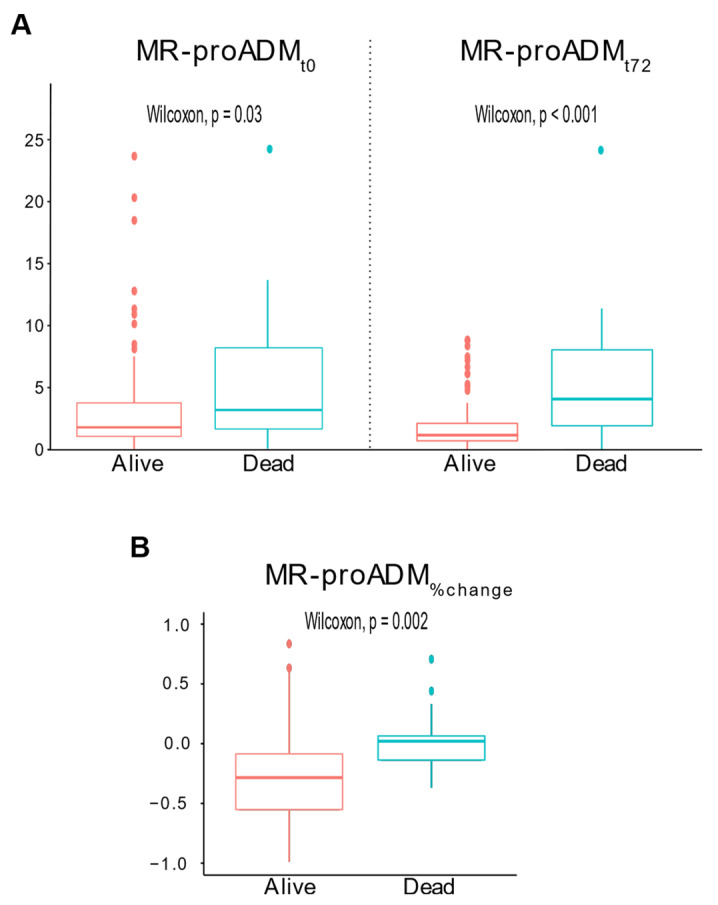
Boxplots of MR-proADM_t0_ and MR-proADM_t72_ for survivors and non-survivors (panel (**A**)); boxplot of MR-proADM_%change_ for survivors and non-survivors (panel (**B**)).

**Figure 2 biomedicines-10-00719-f002:**
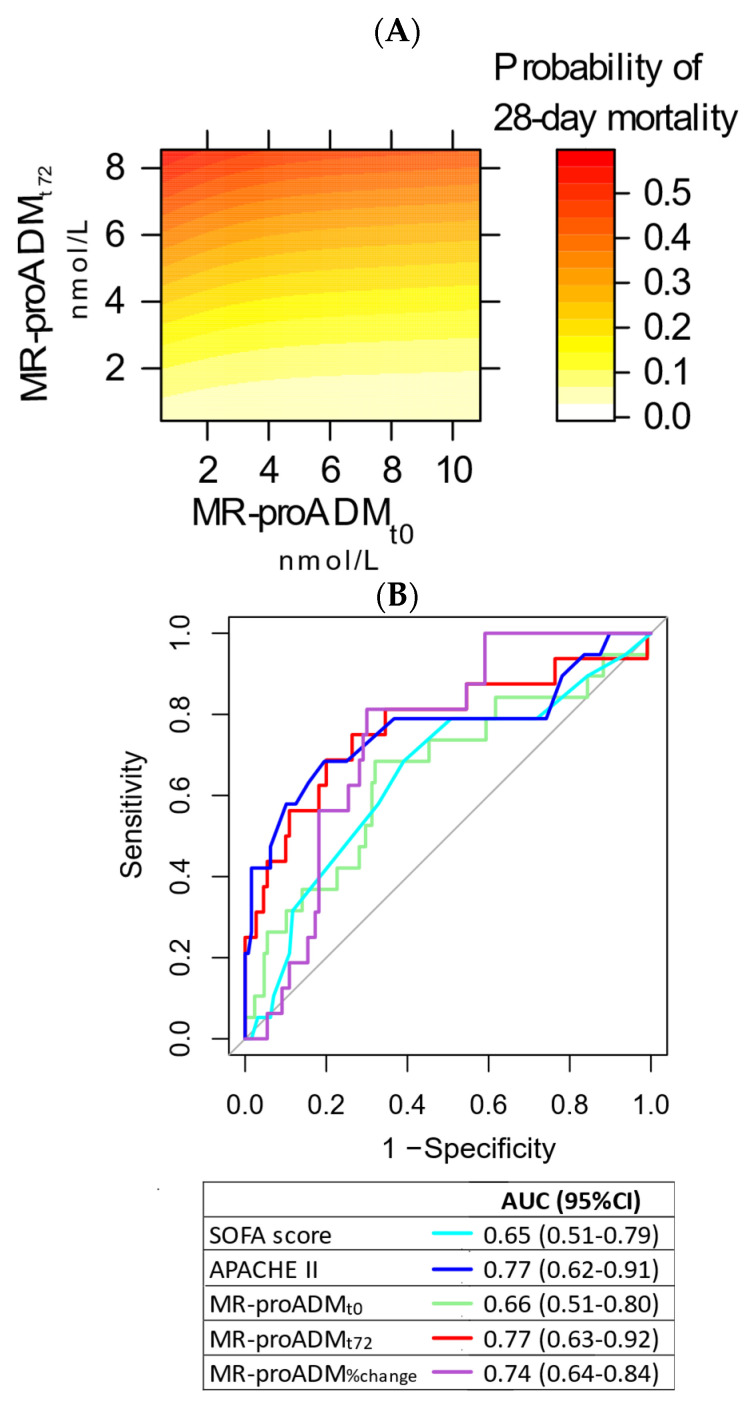
Contour plot showing the interaction of proADM_t0_ and proADM_t72_ in the prediction of 28-day mortality (panel (**A**)). ROC curve analysis for 28-day mortality (panel (**B**)).

**Figure 3 biomedicines-10-00719-f003:**
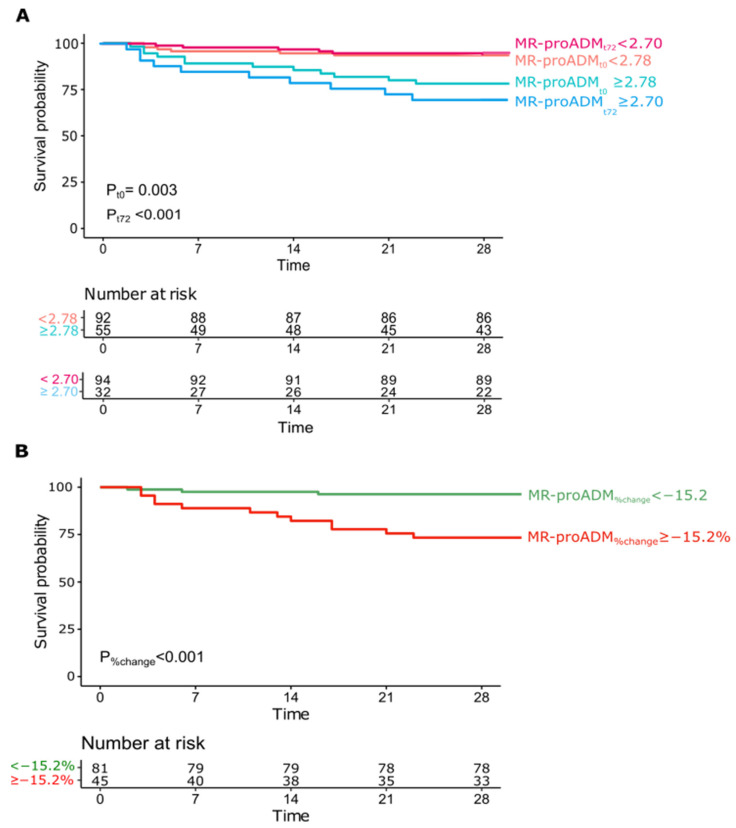
Kaplan–Meier estimator curves for MR-proADM_t0_ ≥ 2.78 and MR-proADM_t72_ ≥ 2.70 (panel (**A**)), and MR-proADM_%change_ ≥ −15.2% (panel (**B**)).

**Figure 4 biomedicines-10-00719-f004:**
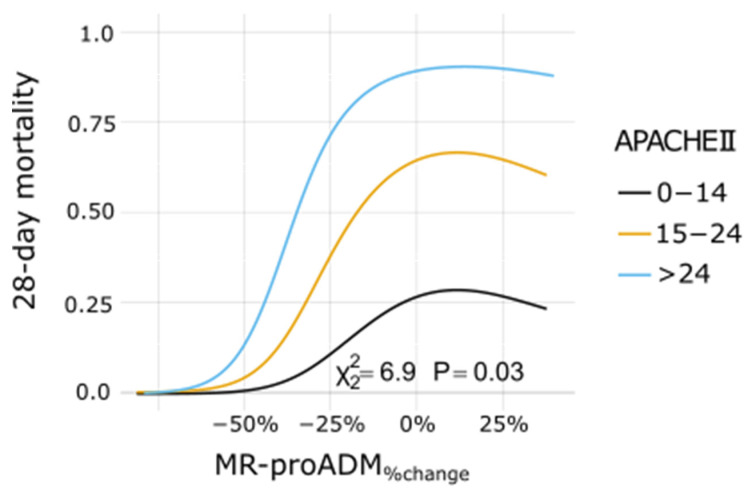
Dose–response plot for 28-day mortality predicted by MR-proADM_%change_ at varying levels of the APACHEII score.

**Table 1 biomedicines-10-00719-t001:** Clinical characteristics of the patients.

	Total (N = 147)	Survivors at 28 Days (N = 128)	Non-Survivors at 28 Days (N = 19)	*p*-Value
Age (years)	69 ± 16	67 ± 16	78 ± 10	0.001
Sex (F)	59 (40.1%)	50 (39.1%)	9 (47.4%)	0.62
**Pre-existing comorbidities**				
Hypertension	89 (60.5%)	75 (58.6%)	14 (73.7%)	0.21
Smoking	31 (21.1%)	25 (19.5%)	6 (31.6%)	0.45
CAD	30 (20.4%)	25 (19.5%)	5 (26.3%)	0.49
Recent (<3 months) hospital admission	31 (21.1%)	28 (21.9%)	3 (15.8%)	0.54
Chronic steroids	32 (21.8%)	27 (21.1%)	5 (26.3%)	0.61
Immunosuppression	22 (15.0%)	18 (14.2%)	4 (23.5%)	0.57
HIV	0 (0%)	-	-	-
Active solid cancer	28 (19.0%)	21 (16.4%)	7 (36.8%)	0.03
Active hematologic malignancy	16 (10.9%)	13 (10.2%)	3 (15.8%)	0.44
Chronic kidney disease	41 (27.9%)	37 (28.9%)	4 (21.1%)	0.59
Diabetes	34 (23.1%)	33 (25.8%)	1 (5.3%)	0.08
COPD	30 (20.4%)	24 (18.8%)	6 (31.6%)	0.20
NYHA IV heart failure	5 (3.4%)	5 (3.9%)	0 (0%)	1.0
Liver cirrhosis	5 (3.4%)	5 (3.9%)	0 (0%)	1.0
Recent (<3 months) trauma	6 (4.1%)	6 (4.7%)	0 (0%)	1.0
Recent (<3 months) burns	0 (0%)	-	-	-
**Vital parameters**				
Blood pressure				
SBP (mmHg)	112 ± 26	114 ± 27	99 ± 19	0.017
DBP (mmHg)	63 ± 15	65 ± 15	56 ± 11	0.005
MAP (mmHg)	80 ± 17	81 ± 18	70 ± 12	0.006
Heart rate (bpm)	99 ± 20	98 ± 19	106 ± 23	0.04
Respiratory rate (bpm)	20 ± 6	19 ± 6	24 ± 7	0.03
SpO2 (%)	93 ± 5	93 ± 5	92 ± 7	0.40
Temperature (°C)	38.0 ± 1.6	38.2 ± 1.2	36.5 ± 2.6	0.001
AVPU scale				
Awake	112 (76.2%)	101 (78.9%)	11 (57.9%)	0.05
Vocal	10 (6.8%)	9 (7.0%)	1 (5.3%)	0.8
Pain	5 (3.4%)	3 (2.3%)	2 (10.5%)	0.13
Unresponsive	20 (13.6%)	15 (11.7%)	5 (26.3%)	0.14
Glasgow Coma Scale (GCS)	13 ± 4	13 ± 4	11 ± 5	0.05
**Laboratory parameters**				
P/F ratio	286 ± 97 Missing: 16 (10.9%)	293 ± 93	238 ± 111	0.06
White blood cells (×10^9^)	14.69 ± 9.40	14.35 ± 9.00	16.92 ± 11.82	0.52
Hemoglobin (g/dL)	12.1 ± 2.4	12.4 ± 2.3	10.5 ± 2.3	0.002
Hematocrit (%)	37.2 ± 7.0	37.7 ± 6.7	33.6 ± 8.0	0.01
Platelets (×10^3^)	214 ± 128	212 ± 125	228 ± 145	0.55
Creatinine (mg/dL)	2.0 ± 1.9	1.91 ± 1.6	2.70 ± 3.22	0.27
Bilirubin (mg/dL)	1.1 ± 0.9	1.1 ± 0.8	1.2 ± 1.0	0.98
**Final diagnosis**				
Sepsis-3, sepsis	109 (74.1%)	98 (85.2%)	11 (57.9%)	0.10
Sepsis-3, septic shock	38 (25.9%)	30 (23.4%)	8 (42.1%)	0.10
**Microbiology**				
Positive cultures	75 (51.0%)	66 (52.4%)	9 (47.4%)	0.68
Gram positive	25 (17.0%)	23 (18.0%)	2 (11.1%)	0.53
Gram negative	29 (19.7%)	25 (19.5%)	4 (22.2%)	1.0
Fungal	3 (2.0%)	3 (2.3%)	0 (0%)	1.0
Multiple microorganisms	17 (11.6%)	15 (11.7%)	2 (11.1%)	1.0
**Origin of infection**				
Lungs	57 (38.8%)	47 (36.7%)	10 (52.6%)	0.18
Urinary tract	35 (23.8%)	33 (25.8%)	2 (10.5%)	0.25
Soft tissue	7 (4.8%)	7 (5.5%)	0 (0%)	0.60
Gastrointestinal	21 (14.3%)	18 (14.1%)	3 (15.8%)	0.74
Joints	5 (3.4%)	4 (3.1%)	1 (5.3%)	0.50
Heart valves	1 (0.7%)	1 (0.8%)	0 (0%)	1.0
Central venous catheter associated/Bacteriemia	9 (6.1%)	8 (6.3%)	1 (5.3%)	1.0
Unknown	15 (10.2%)	13 (10.2%)	2 (10.5%)	1.0
**Treatment**				
Antibiotics				
Any	147 (100%)	128 (100%)	19 (100%)	-
Ceftriaxone	38 (25.9%)	34 (26.6%)	4 (21.1%)	0.78
Piperacillin/Tazobactam	47 (32.0%)	40 (31.3%)	7 (36.8%)	0.61
Carbapenems	46 (31.3%)	40 (31.3%)	6 (31.6%)	1
Vancomicin	26 (17.7%)	21 (16.4%)	5 (26.3%)	0.1
Vasopressors	38 (25.9%)	30 (23.4%)	8 (42.1%)	0.10
Corticosteroids	32 (21.8%)	27 (31.4%)	5 (55.6%)	0.56
Conventional oxygen therapy	28 (19.0%)	18 (14.1%)	10 (52.6%)	<0.001
Non-invasive mechanical ventilation	21 (14.3%)	14 (10.9%)	7 (36.8%)	0.007
Mechanical ventilation	24 (16.3%)	17 (13.3%)	7 (36.8%)	0.02
Renal replacement therapy	15 (10.2%)	11 (8.6%)	4 (21.1%)	0.03
**Biomarkers at ED presentation**				
MR-proADM_t0_ (nmol/L)	1.93 (1.10–4.28)	1.79 (1.07–3.78)	3.19 (1.36–9.37)	0.03
PCT (ng/mL)	2.2 (0.64–12.3) Missing: 8 (5.4%)	2.0 (0.6–11.3)	6.9 (0.8–15.3)	0.31
Lactate (mmol/L)	2.0 (1.1–3.3) Missing: 32 (21.8%)	1.9 (1.1–3.3)	2.7 (1.7–6.3)	0.05
CRP (mg/L)	55 (16–192)	64 (17–197)	30 (8–98)	0.12
**Biomarkers at 72-h from presentation**				
MR-proADM_t72h_ (nmol/L)	1.41 (0.84–2.89) Missing: 21 (14.3%)	1.29 (0.82–2.30)	4.31 (2.00–8.86)	<0.001
PCT (ng/mL)	2.3 (0.58–12.2) Missing: 56 (38.1%)	1.9 (0.6–11.5)	19.3 (8.1–26.3)	0.14
Lactate (mmol/L)	1.4 (0.9–1.8) Missing: 123 (83.7%)	1.0 (0.9–1.6)	2.3 (1.6–5.7)	0.04
CRP (mg/L)	114 (51–194) Missing: 55 (37.4%)	111 (54–190)	196 (33–279)	0.44
MR-proADM_%change_	−26% [−53–(−3%)] Missing: 21 (14.3%)	−28% [−55–(−8%)]	2% (−14–7%)	0.002
**Severity scores at ED presentation**				
SOFA score	4 (2–6)	4 (2–6)	6 (4–7)	0.03
APACHE II	21 (15–26)	20 (15–24)	29 (23–37)	<0.001

CAD = coronary artery disease, HIV = human immunodeficiency virus, COPD = chronic obstructive pulmonary disease, NYHA = New York Heart Association classification, SBP = systolic blood pressure, DBP = diastolic blood pressure, MAP = mean arterial pressure, SpO2 = peripheral oxygen saturation, P/F ratio = arterial partial oxygen pressure and fraction of inspired oxygen ratio, ED = emergency department, MR-proADM = mid-regional proadrenomedullin, PCT = procalcitonin, CRP = C-reactive protein.

**Table 2 biomedicines-10-00719-t002:** Prognostic performance measures.

Test	Sensitivity (95% CI)	*p*-Value	Specificity (95% CI)	*p*-Value	LR+ (95% CI)	*p*-Value	LR− (95% CI)	*p*-Value
MR-proADM_t0_ ≥ 2.78 nmol/L	81.3% (62.1–100)	-	64.6% (55.6–73.5)	-	2.3 (1.6–3.2)	-	0.3 (0.1–0.8)	-
MR-proADM_t72_ ≥ 2.7 nmol/L	68.8% (46.0–91.5)	0.5	80.0% (72.5–87.5)	<0.001	3.4 (2.1–5.7)	0.05	0.4 (0.2–0.8)	0.4
MR-proADM_%change_ ≥ −15.2%	75.0% (53.8–96.2)	1.0	70.0% (61.4–78.6)	0.5	2.5 (1.7–3.7)	0.8	0.4 (0.2–0.8)	0.8

LR: likelihood ratio.

## Data Availability

Data will be available upon reasonable request.

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
