# Peer review of "Prognostic Value of Mid-Regional Proadrenomedullin Sampled at Presentation and after 72 Hours in Septic Patients Presenting to the Emergency Department: An Observational Two-Center Study"

_biomedicines, 2022, doi:10.3390/biomedicines10030719_

Round 1
Reviewer 1 Report
The authors are to be congratulated with a very good paper.
I have only tiny comments:
I suggest explaining the acronym LR in the legends to Table 2
I also suggest putting the definition of PCT in parenthesis after the acronym in the discussion.
Author Response
Reviewer 1
I suggest explaining the acronym LR in the legends to Table 2
I also suggest putting the definition of PCT in parenthesis after the acronym in the discussion.
Response to reviewer 1:
We made the changes accordingly.
Reviewer 2 Report
Is a study that must be confirmed by other research in the field.It would be interesting to know the costs of such a protocol with MR-proADM.
Author Response
Reviewer 2:
Is a study that must be confirmed by other research in the field.
It would be interesting to know the costs of such a protocol with MR-proADM.
Response to Reviewer 2:
The cost of serial sampling of MR-proADM is an interesting point. We added a comment about it in the Discussion (page 10 lines 36-37).
In our institutions a MR-proADM test costs around 22€, while a procalcitonin test costs around 11€. Therefore, measuring MR-proADM at arrival and after 72 hours would cost around 44€ and measuring procalcitonin every day would cost the same.
Reviewer 3 Report
In this observational two center study authors are evaluating the significance of mid-regional proadrenomedullin sampled at presentation and after 72 hours as a prognostic marker, in septic patients presenting to the Emergency Department. The study is well designed and manuscript is well written.
Following are some comments.
- This is a very nicely executed study design for easy read authors should provide a consort diagram of the study where they can depict the blinding, inclusion exclusion criteria, distribution and outcomes.
- Authors have tabulated a varieties of diseases which apparently lead to sepsis. A subgroup analysis is highly desirable between different groups of patients to dissect out the independent role of those occurrences on the level of MR proADM.
Author Response
Reviewer 3:
In this observational two center study authors are evaluating the significance of mid-regional proadrenomedullin sampled at presentation and after 72 hours as a prognostic marker, in septic patients presenting to the Emergency Department. The study is well designed and manuscript is well written.
Following are some comments.
- This is a very nicely executed study design for easy read authors should provide a consort diagram of the study where they can depict the blinding, inclusion exclusion criteria, distribution and outcomes.
- Authors have tabulated a varieties of diseases which apparently lead to sepsis. A subgroup analysis is highly desirable between different groups of patients to dissect out the independent role of those occurrences on the level of MR proADM.
Response to Revewer 3:
- We added a flow diagram depicting the study flow in Supplementary figure 1. We adapted the CONSORT flow diagram as it was designed for randomized-controlled trials.
- We added a subgroup analysis for the most common infective focus, which in our cohort was the lung (please see page 8 lines 14-22, page 9 lines 20-22, supplementary tables 3 and 6). We did not carried out further subgroup analysis given the small number of patients and events in those subgroups.